# Capture of Circulating Tumour Cell Clusters Using Straight Microfluidic Chips

**DOI:** 10.3390/cancers11010089

**Published:** 2019-01-14

**Authors:** Arutha Kulasinghe, Jian Zhou, Liz Kenny, Ian Papautsky, Chamindie Punyadeera

**Affiliations:** 1The School of Biomedical Sciences, Institute of Health and Biomedical Innovation, Queensland University of Technology, Kelvin Grove, QLD 4059, Australia; arutha.kulasinghe@qut.edu.au; 2Translational Research Institute, Brisbane, QLD 4102, Australia; 3Department of Bioengineering, University of Illinois at Chicago, Chicago, IL 60607, USA; jzhou88@uic.edu (J.Z.); papauts@uic.edu (I.P.); 4University of Illinois Cancer Center, Chicago, IL 60612, USA; 5School of Medicine, University of Queensland, QLD 4029, Australia; lizkenny@bigpond.net.au; 6Central Integrated Regional Cancer Services, Royal Brisbane and Women’s Hospital, Queensland Health, QLD 4029, Australia

**Keywords:** circulating tumour cells (CTCs), CTC clusters, circulating tumour microemboli, head and neck cancers, microfluidic technology

## Abstract

Circulating tumour cells (CTCs) are the metastatic precursors to distant disease in head and neck cancers (HNCs). Whilst the prognostic and predictive value of single CTCs have been well documented, the role of CTC clusters, which potentially have a higher metastatic capacity are limited. In this study, the authors used a novel straight microfluidic chip to focus and capture CTCs. The chip offers high cell recoveries with clinically relevant numbers (10–500 cells/mL) without the need for further purification. Single CTCs were identified in 10/21 patient samples (range 2–24 CTCs/mL), CTC clusters in 9/21 patient samples (range 1–6 CTC clusters/mL) and circulating tumour microemboli (CTM) in 2/21 samples. This study demonstrated that CTC clusters contain EGFR amplified single CTCs within the cluster volume. This novel microfluidic chip demonstrates the efficient sorting and preservation of single CTCs, CTC clusters and CTMs. The authors intend to expand this study to a larger cohort to determine the clinical implication of the CTC subsets in HNC.

## 1. Introduction

Cancer metastasis remains the major cause of cancer-related deaths, yet little is known about the transient cell populations responsible for the formation of distant disease. Circulating tumour cells (CTCs) were first described in 1869 by an Australian physician, Thomas Ashworth, who discovered cells similar to those in the primary tumour itself in the blood of a patient [1]. In 1954, Watanabe described the role of CTC clusters leading to metastases more readily than single CTCs [2]. Since these findings, the field has gained significant traction in technologies for isolating rare cancer cells from patient bloods [3,4,5]. To this end, the first FDA-approved CellSearch platform (Menarini Silicon Biosystems, Bologna, Italy) was developed. This technology pre-selects for epithelial cell adhesion molecule expression (EpCAM) on cells before characterising cells for pan-cytokeratin, CD45 and DAPI [6]. Cells are further morphologically determined to be CTCs by having a high nuclear to cytosplasmic ratio. The CellSearch technology demonstrated that EpCAM + CTCs were prognostic in a number of tumour types [6]. However, the reliance on a single marker for CTC isolation showed limitations, especially with the presence of mesenchymally shifted/stem-like CTC phenotypes [7,8]. CTCs have shown to have extensive heterogeneity mirroring their journey from the primary tumour to the distant site [9].

To facilitate the capture of a greater proportion of CTCs including multicellular tumour aggregates such as circulating tumour microemboli (CTM), label-free technologies were developed [10,11,12,13,14,15]. Whilst the role of CTMs hasn’t been fully understood, the presence of CTMs has been documented to be a predictor of shorter survival [15,16]. Microfluidic platforms have dominated the recent expansion of CTC technologies due to their features which include high capture efficiencies, high system throughputs and low sample volumes [5,11,17,18]. In addition, advances in microfluidic chip technologies have enabled functional analysis such as CTC expansion in vitro [19,20]. Studies have shown that CTC clusters have the capacity to traverse narrow capillaries in single file and retain their cluster properties upon reaching wider blood vessels [21,22]. Moreover, in patients where CTC clusters have been found, they have a higher propensity to metastasize to distant organs and consequently, poorer overall survival compared to patients with single/no CTCs [23].

In this study, we investigated the presence of CTCs and CTMs in head and neck cancers (HNC) using a novel straight microfluidic chip technology which exploits size-based differences between CTCs and haematopoietic cells. The design of the biochip is described in Zhou et al. [24]. The microchip is designed based on inertial migration of cells in a straight microchannel. According to the two-stage migration model of inertial microfluidics [13], cells flowing inside a straight low-aspect-ratio microchannel will laterally migrate to their two equilibrium positions at centreline near top and bottom walls. Due to strongly size-dependent inertial forces dominating the motion of cells, larger cells or clusters migrate much faster into their equilibrium positions, leaving smaller cells unfocused and randomly distributed throughout the channel cross-section (Figure 1A). Such differential migration of cells based on their diameter offers unparalleled convenience of label-free cell separation [14,25,26].

Using the chip, we demonstrated efficient capture of HNC CTCs and CTMs. The chip had high cell recoveries with clinically relevant numbers (10–500 cells/mL) without the need for further purification. CTCs were characterized molecularly for EGFR and found to be commonly amplified. This study demonstrates a one-step enrichment protocol for the capture of HNC CTCs.

## 2. Results

### 2.1. Microfluidic Chip Design

In this work, we used a multi-flow configuration in a straight microchannel for the separation and capture of individual CTCs and their clusters. The channel was 150 µm in width (*w*) and 50 µm in height (*h*), with two inputs and two outputs. Instead of a single sample flow stream, we introduced buffer (PBS) at the inner inlet (I_b_) and sample at the outer inlet (I_s_). Such flow configuration forms two sample flows sandwiching a buffer flow in the middle of the microchannel (Figure 1B). Under influence of inertial forces, cells migrate laterally away from the sample streams into the clean buffer stream. Due to the strong size-dependence of inertial migration, we can precisely control the channel length (i.e., migration time) to select larger CTCs and clusters, leading to separation from white blood cells (WBCs). The former are collected from the inner outlet (IO) and the waste cells exit the channel from the outer outlet (OO). While in this study we used blood samples with Red blood cells (RBCs) lysed, the multi-flow configuration is also capable of processing untreated whole blood which has been demonstrated in our recent work [27].

Since RBCs were lysed, we set the cut-off size of our channel to 14 µm in order to differentiate CTCs and clusters from WBCs. Based on the two-stage model of inertial migration [13] and the critical role of outlet resistance in flow fractionation [28], the cut-off size is determined by the channel length and by the fluid resistance ratio of the outlets [28]. Setting identical channel branches after the bifurcation, we equalized the fluid resistance of each channel branch. In this case, the migration distance for cells was roughly one third of the channel width (~50 µm) in horizontal direction (Stage II migration). The migration distance in vertical direction (Stage I migration) was about one half of channel height (~25 µm). These are similar to the conditions of cell migration in a 100 µm × 50 µm channel amid the difference in channel aspect ratio. Since the focusing length (*L*) inversely scales with the square of cell size (*L* ∝ *a*^−2^), we can estimate the channel length based on the focusing length of 20 µm cells [29] at Reynolds number *Re* = 50 in our prior work. The estimate for 14 µm cut-off size is *L =* 17 mm. Thus, we set the length to 20 mm to accommodate differences in channel aspect ratio and variation of operational flow rate. The size distribution of separated cells in (Figure 1C) confirms the cut-off size.

### 2.2. Characterization of Chip with Head and Neck Cancer Cell Lines

Three HNC cell lines, which have a mean diameter above 15 µm, were used for spike-in and recovery experiments. The numbers of cells used for spike-in experiments were relatively low to reflect clinically relevant CTC counts which are typically found in HNC (10–500 cells total). The percentage recovery for cell counts between 50–500 cells ranged from 70–96%. For spike-in experiments of 10 cells, the percentage recovery ranged from 40–80% (Figure 2A). The straight chip allows for a higher percentage recovery without a need for multiple rounds of re-processing which can be required with other technologies [12,30,31]. The staining specificity of the CellSearch antibody cocktail is shown in (Figure 2B) where cancer cells FaDu (ATCC^®^HTB43^TM^, Manassas, VA, USA) are detectable in a background of WBCs.

### 2.3. Characterization of the Chip Enriched Head and Neck Cancer Patient Blood Samples

A total of *n* = 21 HNC patients ranging from early to advanced disease (Stage I (*n* = 4), Stage II (*n* = 4), Stage III (*n* = 8), Stage IV (*n* = 5)) were recruited for this study (Table 1). The patient cohort consisted of 81% males and 19% females, with an average age of 60.3 years (range 47–85 years). 14 of the 21 HNC patients were human papillomavirus (HPV) positive. In 10/21 patients, single CTCs were detectable (Range of 2–24 CTCs/mL; 2/4 Stage I, 1/4 Stage II, 4/8 Stage III, 3/5 Stage IV). CTC clusters were found in 9/21 patients (Range 1–6 CTC clusters/mL; 1/4 Stage I, 2/4 Stage II, 2/8 Stage III, 4/5 Stage IV) (Figure 3A) where the size of CTC clusters ranged from 80.26 µm–353.10 µm (Figure 3B). Notably, single CTCs and CTC clusters (Figure 4A) were present at all stages of disease and the presence of CTC clusters increased with an advancement in staging, indicative of a higher burden of disease [32].

In two stage IV HNC patients, CTMs were detected (Range 1–5 CTMs/mL) consisting of CTCs and WBCs within the aggregate (Figure 4B). The size of CTMs ranged from 133.30 µm–340.54 µm which was comparable to CTC clusters (Figure 3B). Models have been proposed for CTMs which include tumour cells, endothelial cells, platelets, fibroblasts, leukocytes and pericytes within the cellular aggregate [33].

Single CTCs and CTC clusters were found in five patients and either single/clusters of CTCs were found in 14/21 patients (66.7%). In five CTC positive samples, EGFR DNA FISH was performed showing EGFR gene amplification in all five CTC positive samples (minimum of at least one individual CTC/CTC clusters showing EGFR amplification) (Figure 5A–C). No CTC-like events were observed in the five normal healthy volunteer samples.

## 3. Discussion

Microfluidic technology, such as the straight chip allows for a simple to use, size based separation of CTCs, with high recovery efficiencies and low background cell contamination (white blood cells). This holds remarkable potential in identifying CTCs in a single enrichment processing step without the need for further purification. Current CTC enrichment technologies which enrich for cancer cells using surface marker expression (positive selection), or the removal of non-cancer cells (negative selection) are poor in discriminating between individual CTCs and CTC clusters. By using negative enrichment technologies (e.g., CD45 depletion that target WBCs), there remains a potential risk in losing CTMs which may have CD45 positive cells at the periphery of the cellular aggregate.

There have been a few early reports of CTC clusters in HNC [34,35,36] with most cases in advanced stages of disease. Interestingly, in this study, CTC clusters were found at each stage of disease. Whether this is because CTC clusters were present in early stage HNC, or the patients had underlying advanced locoregional disease which wasn’t obvious at diagnosis remains to be ascertained. The CTC clusters found in this study appeared to be more preserved compared to other microfluidic technologies, reflecting the gentle processing of the straight chip to maintain CTCs [12,36]. The 3D volumetric depiction of CTC clusters as in (Figure 4), showed better resolution of the individual CTCs forming the cluster. There is now growing evidence documenting the presence of CTC clusters in metastatic tumours, including aggressive forms of brain cancers [21,22,23,33,37,38]. CTC clusters have a shorter half-life in blood, with a higher propensity to metastasize to distant organs compared to single CTCs [23]. These clusters have specialized microenvironments which are not simply a result of cellular aggregation [39]. In breast cancer, plakoglobin and keratin 14 have been reported to be associated with desmosomes and hemidesmosomes and crucial for CTC cluster formation. The inhibition of these has resulted in disruption of cluster formation and distant metastases [23,40]. In a mouse model of breast cancer metastasis, injection of urokinase-type plasminogen activator in the host animal was effective at preventing CTC cluster assembly and improved the host survival by 20% compared to control animals [41]. CTC cluster studies have shown that there remains vast heterogeneity even within CTC clusters, representing complex compositions and crosstalk between cells [42]. Therefore, it remains critical to isolate and profile CTC clusters to better understand the role of these cells in tumour metastasis. In a number of studies, CTC clusters have been reported to be independent prognostic markers for poorer patient outcomes [36,42,43]. The presence of immune cells within the CTM may provide insights into how CTMs evade the immune system and contribute to the survival and metastatic advantages [42].

Capturing and analysing CTCs for genetic and molecular alterations allows for the identification of potential targeted therapies which may benefit the patient (e.g., EGFR targeted therapies in HNC). This study showed that HNC CTCs including CTC clusters had EGFR amplified cells which is a common event in HNC [44]. Other common amplifications in HNC are FGFR1 (10%), CCND1 (31%) MYC (14%) and PIK3CA (34%) [44]. The authors believe this to one of the first studies to document the presence of EGFR amplified cells within CTC clusters found in HNC and seek to determine the frequency of other common mutation in a follow on study.

Whilst this is the first generation of the straight microfluidic chip, which can capture single and clusters of CTCs, there is an opportunity to increase the size-based focussing of cells to exclusively capture CTC clusters/microemboli (e.g., above 30 µm). Moreover, the chip has the ability to work with lower blood dilutions and potential to process undiluted whole blood. A limitation of this chip is that small CTCs which have a comparable size to white blood cells are unlikely to be captured using size based exclusion technologies [45]. Furthermore, head to head comparisons of this chip to the FDA-approved CellSearch are warranted to determine the CTC populations captured by each technology and their role in metastasis.

## 4. Materials and Methods

### 4.1. Chip Fabrication

Microchannels were fabricated via standard soft photolithography. We used dry film resist instead of conventional liquid SU-8 photoresist, which simplifies the fabrication process and provides better uniformity in terms of channel height. Briefly, dry film resist (ADEX 50, DJ MicroLaminates Inc., Boston, MA, USA) was used to pattern microchannels on a 3′′ silicon wafer by photolithography. Polydimethylsiloxane (PDMS, Dow Corning^®,^ Midland, MI, USA) was casted on the wafer and peeled after 6-h curing on 65 °C hotplate. Replicated straight channels (150 μm wide, 50 μm high, and 20 mm long) in PDMS were bonded to 1′′ × 3′′ glass slides (Fisher Scientific, Hampton, NH, USA) using surface plasma treatment (PE-50, PlasmaEtch Inc., Carson City, NV, USA). The inlet and outlet ports were punched manually using stainless flat head needles before the channels were sealed onto glass slides.

### 4.2. Chip Operation and Workflow

Sample and buffer solutions were injected into the PDMS device with two syringe pumps (Fusion 200 Touch, Chemyx, Stafford, TX, USA) to sustain stable flow rate. The loaded syringes were connected to 1/16′′ Tygon^®^ tubings (IDEX Health & Science LLC, Oak Harbor, WA, USA) using proper fittings and then secured to the device inlets. The sample and PBS loaded syringes were connected to the outer inlet (I_s_) and inner inlet (I_b_) of the microchip using Tygon^®^ tubing, respectively. The two outlets of the microchip were connected to two separation containers via Tygon^®^ tubings for target cells (CTCs) and waste cells (WBCs). The flow rates for sample and PBS solution were 100 and 200 µL/min respectively.

### 4.3. Chip Characterization Using Cell Lines

The performance of the straight microfluidic chip was assessed using HNC cell lines Fadu (ATCC^®^HTB43^TM^), 2A3 (ATCC^®^CRL3212^TM^) and CAL27 (CRL^®^2095 ^TM^) sourced from the American Type Culture Collection (ATCC, Manassas, VA, USA). Cells were cultured under standard conditions in humidified incubators at 37 °C, 5% CO_2_ in RPMI 1640-Glutamax (Life Technologies, Inc., Carlsbad, CA, USA) supplemented with 10% foetal bovine serum (FBS) and 1% Penicillin/Streptomycin. Cells were lifted using Tryple Express reagent (ThermoFisher Scientific, Waltham, MA, USA), resuspended in media and gently mixed on a shaker prior to experiments. Cell lines for spiking and recovery experiments were labelled with CellTracker™ (Life Technologies, Inc., Carlsbad, CA, USA) as per manufacturer’s instructions. Cell line authenticity was confirmed by short tandom repeat (STR) profiling with GenePrint^®^ 10 system (Promega, Madison, WI, USA) by the Genomic Research Centre (/IHBI/QUT). Unlabelled spike-in cells were stained with the CellSearch (Menarini Silicon Biosystems, Bologna, Italy) antibody cocktail targeting cytokeratin-8,18,19, CD45 and DAPI.

### 4.4. Patient Recruitment

This study was conducted at the Royal Brisbane and Women’s Hospital in Brisbane. Ethics approval was obtained from the Metro South Health district Human Research Ethics Committee in accordance with the National Health and Medical Research Councils guidelines and Declaration of Helsinki. Human Ethics Approval number (HREC/12/QPAH/381). This study also has institutional approval from the Queensland University of Technology Human Ethics Committee: Approval number (1400000617). Following written informed consent, 10 mL of blood was collected in K2E EDTA vacutainers from a total of 26 participants (*n* = 21 HNC patients and *n* = 5 healthy volunteers). All HNC patients were treatment naïve at the time of blood collection.

### 4.5. Enrichment and Phenotyping of CTCs/CTMs

Blood samples collected from the participants were lysed using RBC lysis buffer (Astral Scientifix, Taren Point, NSW, Australia) and centrifuged at 500× *g* for 15 min. The cell pellet was resuspended in 10 mL of 1× PBS solution. The pre-diluted sample was loaded onto a 10 mL Terumo syringe and passed through the inlet of the chip at 0.1 mL/min whilst the sheath buffer (1× PBS) was pumped through at 0.2 mL/min. The CTC sample outlet was collected in a sterile 15 mL falcon tube (Corning, New York, NY, USA) and centrifuged at 300× *g* for 5 min to obtain a pellet. The cell pellet was resuspended in 4% paraformaldehyde and cytocentrifuged (Cytospin™ 4, Waltham, MA, USA) at 800 rpm for 5 min onto glass slides for phenotyping. Cells were permeabilised in 0.2% (*v*/*v*) Triton X-100 for 10 min at room temperature, followed by blocking with 10% FBS (Invitrogen, Carlsbad, CA, USA) for 1 h at room temperature in a humid chamber. Slides were incubated with the CellSearch antibody cocktail including DAPI (Menarini Silicon Biosystems, Bologna, Italy) overnight at 4 °C. Slides were washed 3 times in PBS, allowed to air dry and mounted with antifade prolong gold (ThermoFisher Scientific, Carlsbad, CA, USA). The cytospots were coverslipped and imaged on the Axio Imager 2 (Zeiss, Oberkochen, Germany). CTCs were visually inspected and classified as CTCs after meeting the following criteria: (1) cytokeratin-8,18,19 positive (2) CD45 negative (3) high nucleus to cytoplasmic ratio (4) morphologically larger than background leukocytes (5) intact nuclei. CTC clusters were reported as CTCs in close proximity and CTM when CTC clusters included leukocytes (CD45 positive cells). The results were reported as the number of CTCs per mL whole blood. CTC positive samples were further investigated using EGFR DNA FISH probes as previously described [12]. Briefly, slides were dehydrated using an ethanol series (70%, 85%, 96%) and treated with 4 mg/mL RNase (Sigma, St Louis, MO, USA) and the DNA FISH carried out using the EGFR/CEP-7 FISH probe mix (DakoCytomation, Glostrup, Denmark) and counterstained with DAPI. The slides were imaged on the Zeiss Axio Imager Z2 microscope. EGFR status was scored as the ratio of the number of EGFR signals (red) to CEP-7 (green). An increase in copy number of the EGFR gene is represented by a higher number of red to green signals. A ratio of EGFR:CEP-7 of >3 shows an EGFR amplified cell whereas a ratio of <2 is non-amplified.

### 4.6. Cell Imaging

Scanning for CTCs was performed on the Zeiss Axio Z2 microscope and sequential images were captured after fluorescent staining. A multi-exposure protocol was used to detect the signals. 3D volumetric analysis was performed using z-stacking (distance between z-stacks ranged from 0.1 µm–1 µm). The Zen software was used to interrogate the 2D and 3D images and constrained iterative algorithms used for image deconvolution.

## 5. Conclusions

In this study, the authors validated the straight microfluidic chip for separating single CTCs and CTC clusters from HNC patient blood samples. This is one of the first studies to demonstrate CTMs in HNC patients’ blood samples. It is important to understand the biological roles of these cell populations in the context of metastasis. The study demonstrated that EGFR amplified CTCs are present in CTC clusters confirming that these cells are of HNC tumour origin. The authors intend to expand this study to a larger HNC patient cohort and compare the CTCs findings with patient survival and treatment outcomes.

## Figures and Tables

**Figure 1 cancers-11-00089-f001:**
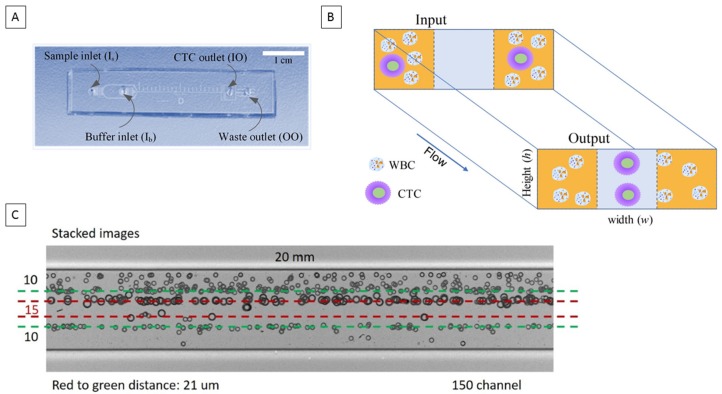
(**A**) Image of the straight microfluidic chip with descriptors for inlets; Sample Inlet (I_s_), Buffer Inlet (I_b_)/outlets; circulating tumour cells (CTC) outlet (IO) and waste outlet (OO). Scale bar represents 1 cm. (**B**) A schematic representing how CTCs (single CTCs and CTC clusters) are focused to the centre of the channel. (**C**) Stacked image showing the focussing of 10 and 15 µm beads through the straight chip. 150 channel represents the channel width in µm. WBC: White blood cells.

**Figure 2 cancers-11-00089-f002:**
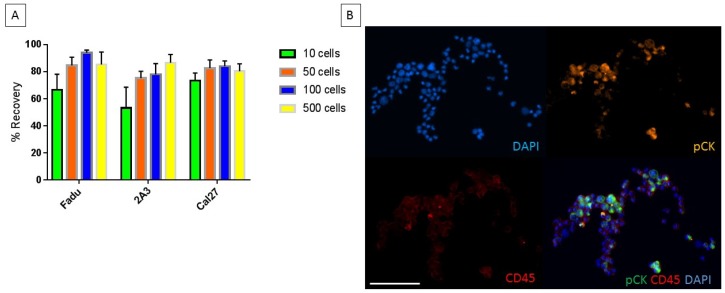
(**A**) The percentage recovery of head and neck cancer (HNC) cell lines spiked into 10 × diluted blood and recovered through the straight microfluidic chip. Green (10 cells), orange (50 cells), blue (100 cells) and yellow (500 cells). (**B**) The multicolour stain for pan-cytokeratin (orange), CD45 (red) and nuclear stain DAPI (blue) of HNC cells spiked and recovered through the straight chip. Scale Bar: 100 µm.

**Figure 3 cancers-11-00089-f003:**
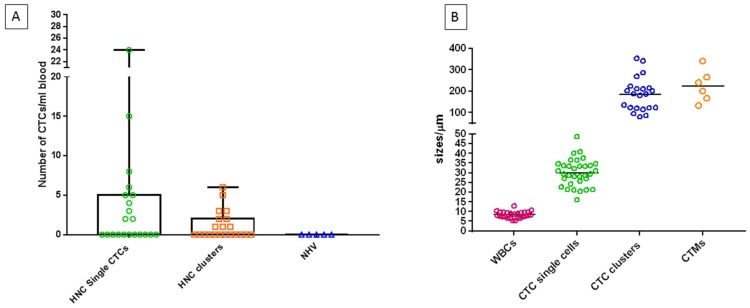
(**A**) The number of CTCs/CTC clusters detected per ml mL blood from HNC patients and normal healthy volunteers (NHV) processed through the straight microfluidic chip. The distribution of single and clustered CTCs/mL is shown and the presence of no CTC-like events in the NHV. (**B**) The size distribution of white blood cells (WBCs), single CTCs and CTC-clusters.

**Figure 4 cancers-11-00089-f004:**
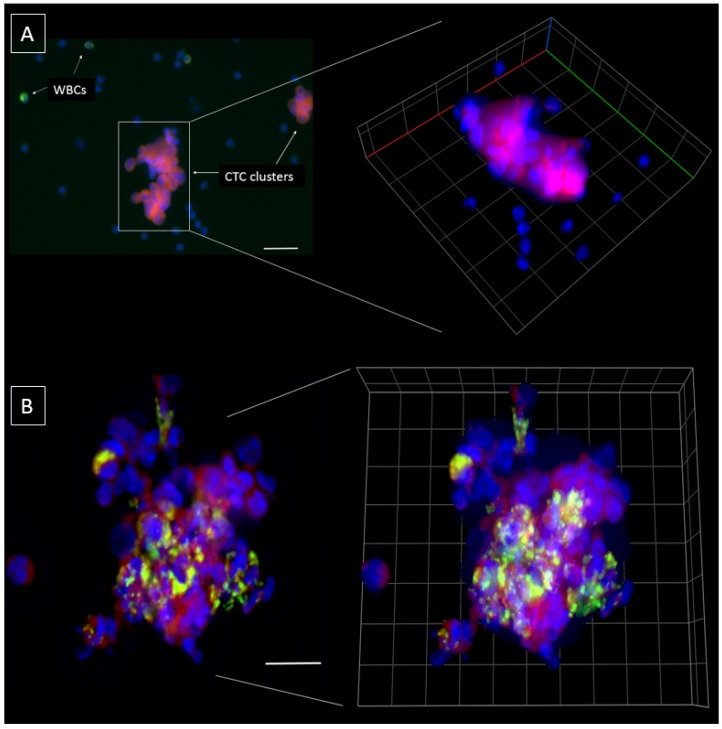
(**A**) The capture of multicellular CTC clusters in the blood of a head and neck cancer patient and (right) 3D volumetric z-stack. (**B**) Multicellular circulating tumour microemboli consisting of CTCs and leukocytes within the aggregate and (right) 3D volumetric z-stack. Cells stained with pan-cytokeratin-8,18,19 (red), CD45 (green) and DAPI (blue). Scale bar represents 50 µm.

**Figure 5 cancers-11-00089-f005:**
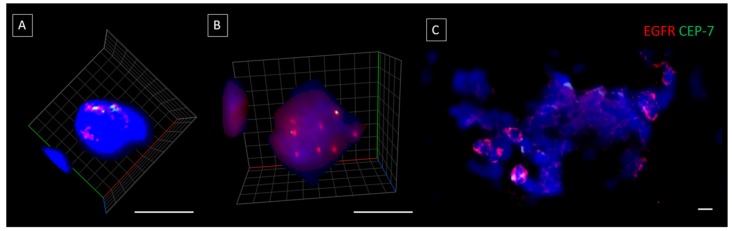
(**A**,**B**) 3D EGFR DNA FISH of single CTCs showing the number of EGFR signals (red) to CEP-7 (green). (**C**) EGFR DNA FISH of a CTC cluster showing a number of EGFR amplified cells. The ratio of EGFR:CEP-7 shows EGFR amplification in the CTCs. Scale bar represented 10 µm.

**Table 1 cancers-11-00089-t001:** Clinicopathological findings for the head and neck cancer cohort (*n* = 21) and CTC findings (single/clusters). HPV: Human papillomavirus (HPV).

Head and Neck Cancer	*n*
Total	21
Gender	
Male	17
Female	4
Age, y	
<60	12
>60	9
Tumour Type	
Oral Cavity	12
Oropharynx	9
Tumour Stage	
I	4
II	4
III	8
IV	5
HPV Status (p16 immunostainings)	
HPV-positive	14
HPV-negative	7
CTC findings/mL blood	
CTC+ (single cells)	10 (Range 2–24)
CTC clusters	9 (Range 1–6)
Single CTCs and clusters	5
CTC (single or cluster)+	14/21 (66.7%)

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
