# Peer review of "Capture of Circulating Tumour Cell Clusters Using Straight Microfluidic Chips"

_cancers, 2019, doi:10.3390/cancers11010089_

Round 1
Reviewer 1 Report
The authors studied single CTC, CTC clusters and CTMs in head and neck cancers by sorting them using a straight microfluidic chip. The device separated the CTCs successfully from the patient samples, and then the EGFR amplified CTCs were identified. However, there are a lot of devices used size-based separation with high recovery rate and purity. The device did not have innovative features for CTC technology, compared to existing devices. In addition, the small population of patient cohort could not show any correlations between the number of CTCs and disease status. Without expanded study, the study provides relatively less impact in this field of study.
1. In this manuscript, the authors distinguished the CTM from CTC clusters as CTM was defined
the clusters including leukocytes. However, the referenced papers (line 50) defined CTM as
CTC aggregates.
2. Figure 1A is not clear enough to see the structure of the chip.
3. In figure 1C, does ‘150 channel’ mean the width of the channel?
4. In the line 110, the authors mentioned the straight chip allows higher recovery rate, but
some other research show that spiral microfluidic chip results higher recovery rate
compared to that of the device suggested in this research.
• Hou, Han Wei, et al. "Isolation and retrieval of circulating tumor cells using centrifugal
forces." Scientific reports 3 (2013): 1259.
• Lin, Eric, et al. "High-throughput microfluidic Labyrinth for the label-free isolation of circulating
tumor cells." Cell systems 5.3 (2017): 295-304.
5. The patient cohort is consisted of 81% males and 19% female. Does HNC incidence rate show
gender difference? Why did you include different portion between male and female?
6. Purity is a critical factor to determine the effectiveness of the CTC isolating devices. Have
you performed the purity test of the device? The staining result (Figure 2B) is not enough to
conclude low background WBC contamination mentioned in the line 157.
7. The labeling with the arrows in figure 4A is too small to read.
8. The CTCs were separated by size-based technology. It enables to detect heterogeneous HNC
cells from blood samples, but they were phenotyped only with EGFR in this research. It
would be better if other genes were studied as well to take advantage of the technology.
Author Response
The authors studied single CTC, CTC clusters and CTMs in head and neck cancers by sorting them using a straight microfluidic chip. The device separated the CTCs successfully from the patient samples, and then the EGFR amplified CTCs were identified. However, there are a lot of devices used size-based separation with high recovery rate and purity. The device did not have innovative features for CTC technology, compared to existing devices. In addition, the small population of patient cohort could not show any correlations between the number of CTCs and disease status. Without expanded study, the study provides relatively less impact in this field of study.
Our manuscript represents a novel CTC biochip which uses microfluidic technology. The chip has the potential to use lower blood dilutions and process undiluted whole blood – now included in lines 197-198 “Moreover, the chip has the ability to work with lower blood dilutions and potential to process undiluted whole blood.” Moreover, in earlier studies which used microfluidic chips focussed on the isolation of individual CTCs and the chip designs have limited capability for size-based isolation of clusters. Futhermore, CTC chip studies generally focus on lung, breast or prostate cancers with head and neck cancers not receiving much attention. This chip has not been reported on previously and is the first paper using this technology for head and neck cancers. The first manuscript describing this biochip in lung cancer has recently been accepted to Microsystems and Nanoengineering (Nature) – Zhou et al., 2019. The authors believe that the sample numbers for the study were sufficient for a preliminary study and envisage expanding this in a subsequent manuscript. The sample numbers are similar to previously published studies using novel biochips (Chudziak et al., Analyst 2016 n=12, Warkiani ME et al., Analyst and Lab on a chip 2014 n=10).
1. In this manuscript, the authors distinguished the CTM from CTC clusters as CTM was defined
the clusters including leukocytes. However, the referenced papers (line 50) defined CTM as
CTC aggregates.
In the literature to date, CTC clusters are generally considered to be a few CTCs in close proximity which travel together in the blood (Hong et al., International Journal of Oncology 2016, Kulasinghe et al., Scientific Reports 2018; Au et al., PNAS 2016). Where the definition changes to become CTMs/CTC aggregates is when they are composed of CTCs including (platelets, stromal and hematopoietic cells).
2. Figure 1A is not clear enough to see the structure of the chip.
The structure of the chip is described in the first manuscript Zhou et al., Microsystems and Nanoengineering (Nature). The purpose of 1A was to give an indication of the microfluidic device.
3. In figure 1C, does ‘150 channel’ mean the width of the channel?
Yes, this has been updated in figure legend 1 (line 71-72) “150 channel represents the channel width in µm”
4. In the line 110, the authors mentioned the straight chip allows higher recovery rate, but
some other research show that spiral microfluidic chip results higher recovery rate
compared to that of the device suggested in this research.
• Hou, Han Wei, et al. "Isolation and retrieval of circulating tumor cells using centrifugal
forces." Scientific reports 3 (2013): 1259.
In the article by Hou et al., high recovery rates are suggested. However, in subsequent publications using the same biochip, the recovery rates are lower (Kulasinghe et al., Scientific Reports 2017 – approximately 25-80% recovery; Aya-Bonilla et al., Oncotarget 2017 - approximately 80%). Furthermore, the biochip described by Hou et al., needs further rounds of enrichment to purify the CTCs from the contaminating background WBCs whereas the straight chip described in this paper does not (Aya Bonilla et al., Oncotarget 2017).
• Lin, Eric, et al. "High-throughput microfluidic Labyrinth for the label-free isolation of circulating
tumor cells." Cell systems 5.3 (2017): 295-304.
In the manuscript by Lin et al., the initial spike in’s were done using high numbers (100000-500000cells/ml) using a number of cell lines. These numbers don’t represent clinically relevant numbers as described in our manuscript (line 74). When lower concentrations of CTCs were used by Lin et al., (100CTCs/ml) this was only performed with one cell line MCF-7.
5. The patient cohort is consisted of 81% males and 19% female. Does HNC incidence rate show
gender difference? Why did you include different portion between male and female?
Head and neck cancers tend to be more common in males than females. It occurs in approximately 25% women which is reflected in our cohort gender distribution.
https://head-neck-cancer.canceraustralia.gov.au/statistics
NCI Head and neck cancers https://www.cancer.gov/types/head-and-neck/head-neck-fact-sheet).
6. Purity is a critical factor to determine the effectiveness of the CTC isolating devices. Have
you performed the purity test of the device? The staining result (Figure 2B) is not enough to
conclude low background WBC contamination mentioned in the line 157.
Figure 2B was included in the manuscript to demonstrate the staining specificity as described in lines 112-113. Purity or the number of contaminating WBCs were not assessed in the study as they were low. In Zhou et al., Zhou et al., Microsystems and Nanoengineering (Nature) – accepted, the device demonstrates that lung cancer CTCs are captured with>87% purity.
7. The labeling with the arrows in figure 4A is too small to read.
Amended figure has been included (Fig 4).
8. The CTCs were separated by size-based technology. It enables to detect heterogeneous HNC
cells from blood samples, but they were phenotyped only with EGFR in this research. It
would be better if other genes were studied as well to take advantage of the technology.
Whilst the authors agree with this sentiment, head and neck cancers are a highly heterogeneous tumour type without any common driver oncogenic events. The reason EGFR amplification was assessed was because this has been shown to be present in approximately 15% of HNC tumours (The Cancer Genome Atlas, TCGA 2015; Comprehensive genomic characterization of head and neck squamous cell carcinomas). Other genes which are amplified include FGFR1 (10%), ERBB2 (5%), DDR2 (3%). Whilst it is desirable to assess other genes, the frequency is low. The reasons for EGFR assessment was to confirm that the CTCs were head and neck cancer in origin.
Reviewer 2 Report
Kulasinghe et al. describe capture and enumeration of CTCs using a microfluidic chip. In particular, they described its utility for capturing CTC clusters and CTMs. This is a well planned and executed study with important implications. I recommend publication based on minor clarifications:
1) It would be good to have a workflow for the protocol including loading on the chip and staining/counting
2) How does cell recovery of single CTCs compare to cellsearch?
3) How does the recovery compare to other microfluidic technologies (e.g. CTC chip)?
4) How were CTC clusters defined? more than 1 or more than 2 CTCs etc.?
5) Line 132. Definition of CTMs/CTC clusters etc. need to come earlier. Perhaps define in the introduction?
Table 2 – Please make CTC (single/cluster) clearer. Perhaps say Single CTCs or clusters?
Author Response
The authors thank the reviewer for the comments and suggestions and believe the manuscript has been strengthened as a result of peer review.
Kulasinghe et al. describe capture and enumeration of CTCs using a microfluidic chip. In particular, they described its utility for capturing CTC clusters and CTMs. This is a well planned and executed study with important implications. I recommend publication based on minor clarifications:
1) It would be good to have a workflow for the protocol including loading on the chip and staining/counting
The authors agree that Figure 1 presents the separation of CTCs clearly. Post enrichment, cells are phenotyped on glass slides. Similar workflows are used for CTC studies (Warkiani ME et al., Nature Protocols 2015, Analyst 2015).
2) How does cell recovery of single CTCs compare to cellsearch?
Head and neck cancers tend to have low EpCAM expression. The CellSearch platform did poorly with head and neck cancer CTC detection in previous studies (Kulasinghe et al., Oncotarget 2016). In cell line studies, EpCAM has shown great variation of EpCAM expression, therefore an epitope independent CTC methodology is preferable (Lindgren et al., Laryngoscope Investig Otolarngol 2017). However, given that the CellSearch is the only FDA-approved platform, the authors have included a statement in the discussion for this “Furthermore, head to head comparisons of this chip to the FDA-approved CellSearch are warranted to determine the CTC populations captured by each technology and their role in metastasis.” (lines 200-202).
3) How does the recovery compare to other microfluidic technologies (e.g. CTC chip)?
It is not clear whether the authors are referring to the original CTC chip or the CTC iChip developed by the Haber and Toner lab. The recoveries are on par with rare cell isolation technologies (Fachin et al., Scientific Reports 2017).
4) How were CTC clusters defined? more than 1 or more than 2 CTCs etc.?
The definition of a CTC cluster varies between studies. However, generally, CTC clusters are two or more cells which travel together in the bloodstream (Hong et al., International Journal of Oncology 2016; Hou et al., Journal of Clinical Oncology 2012).
5) Line 132. Definition of CTMs/CTC clusters etc. need to come earlier. Perhaps define in the introduction?
As the definition of a CTC cluster was described in the original research by Watanabe, this was referenced in lines 35-36. The definition of a CTM was also described in the introduction as cellular aggregates in lines 47-48.
Table 2 – Please make CTC (single/cluster) clearer. Perhaps say Single CTCs or clusters?
Changed to single or cluster
Reviewer 3 Report
General comment
Herein, Kulasinghe and coll. show their results about a microfluidic chip for detecting CTCs and CTC clusters, applied to head and neck cancer.
The study offers the proof of principle that the proposed microfluidic chip can enrich CTCs on the basis of their diameter (physical criteria of selection); authors provide also the proof that the isolated CTCs are of tumour origin, based on FISH analysis for EGFR mutation.
In the opinion of this reviewer, the paper requires minor changes and some statistic to improve clarity and to support soundness of their findings.
Specific comments
Line 20 (Abstract): the authors should precise what does it mean “clinically relevant numbers…”. Their paper does not discuss association between level of CTCs and disease outcome…
Line 22 and 25 (Abstract): In Material and Methods (line 256-258) the authors precise that CTC clusters are aggregate of CTCs (how many?), whereas CTMs include CTCs and CD45+ lymphocytes. However, here and elsewhere in the manuscript, CTC clusters and CTMs are used almost as synonymous. Since CTC clusters and CTMs could have different functional significance on metastatic potential of CTCs, this is misleading and should be emended everywhere in the manuscript.
Line 42: A size larger than Lymphocytes is not included in the CellSearch criteria.
Line 44 and 45: The impact on metastasis of epithelial to mesenchymal transition (EMT) is matter of debate, and deep discussion about this theme is out of the purposes of a research article. However, a note of caution should be included here, since it has been recently reported that EpCAM-positive but not EpCAM-negative CTCs are associated with patients’ outcome (de Wit S, Sci Rep. 2015; de Wit S., Oncotarget 2018), an association that is mandatory for novel assays intended for clinical use. In the opinion of this reviewer the main value of the proposed assay is that it allow enriching unlabelled CTCs.
Line 74 (and line 108): The authors should clarify what they intend for “clinically relevant numbers” (see above, please, line 20). Since previously reported data in head and neck cancer have been obtained by different technologies, include, please, the reference(s) and specify, please, the technology used in that case.
Line 75: Amend, please, “phenotyped molecularly” with “characterized molecularly”.
Line 86: Since the Material and methods section is at the end of the paper, I suggest inserting here, the definition of CTC clusters and CTMs.
Line 91: Haematology atlases report the neutrophils (75% of WBC) fairly uniform in size with a diameter between 12 and 15 μm. Why a diameter of 14 μm should discriminate CTCs and CTMs from WBC?
Line 127 and line 132: How many CTCs are included in CTC clusters and in CTMs? Perform, please, a statistics to compare size of clusters and CTMs (I suggest using non-parametric tests).
Line 147: Report, please, EGFR status in matched primary tumours, if any data is available. In any case, the authors should briefly comment the translational value of this finding in CTCs.
Line 169: On what basis the authors state that their microchip preserves CTC clusters better than other technologies? Include here, please, some objective parameters.
Line 275: See, please, above (line 22 and 25).

Author Response
General comment
Herein, Kulasinghe and coll. show their results about a microfluidic chip for detecting CTCs and CTC clusters, applied to head and neck cancer.
The study offers the proof of principle that the proposed microfluidic chip can enrich CTCs on the basis of their diameter (physical criteria of selection); authors provide also the proof that the isolated CTCs are of tumour origin, based on FISH analysis for EGFR mutation.
In the opinion of this reviewer, the paper requires minor changes and some statistic to improve clarity and to support soundness of their findings.
Specific comments
Line 20 (Abstract): the authors should precise what does it mean “clinically relevant numbers…”. Their paper does not discuss association between level of CTCs and disease outcome
In testing CTC enrichment technologies, a number of publications use high cancer cell line spike in numbers (eg 1000’s of cells) to perform recovery studies. Clinically however, patients present with low CTC counts. Therefore, it is desirable to test CTC platforms using ‘clinically-relevant numbers” of spike-in cells. In our study, we demonstrated this using 10-500 cells/ml. Head and neck cancers typically present with low CTC counts (under 50CTCs/ml – Kulasinghe et al., International Journal of Cancer 2015). Whilst the disease outcomes are desirable to be found on, it is outside the scope of this study and would require a longer patient follow up (typically 1-2 years for head and neck cancer where metastasis would be evident). The authors envisage performing a longitudinal study in the future.
Line 22 and 25 (Abstract): In Material and Methods (line 256-258) the authors precise that CTC clusters are aggregate of CTCs (how many?), whereas CTMs include CTCs and CD45+ lymphocytes. However, here and elsewhere in the manuscript, CTC clusters and CTMs are used almost as synonymous. Since CTC clusters and CTMs could have different functional significance on metastatic potential of CTCs, this is misleading and should be emended everywhere in the manuscript.
The definition of a CTC cluster is not well defined in the field. It is generally termed when two or more CTCs are found to be travelling together in the blood (Hong et al., International Journal of Oncology 2016, Kulasinghe et al., Scientific Reports 2018; Au et al., PNAS 2016). When other cellular components are found to be involved with the CTC cluster it is termed a CTM.
Line 42: A size larger than Lymphocytes is not included in the CellSearch criteria.
This statement has been removed.
Line 44 and 45: The impact on metastasis of epithelial to mesenchymal transition (EMT) is matter of debate, and deep discussion about this theme is out of the purposes of a research article. However, a note of caution should be included here, since it has been recently reported that EpCAM-positive but not EpCAM-negative CTCs are associated with patients’ outcome (de Wit S, Sci Rep. 2015; de Wit S., Oncotarget 2018), an association that is mandatory for novel assays intended for clinical use. In the opinion of this reviewer the main value of the proposed assay is that it allow enriching unlabelled CTCs.
The authors agree with the reviewer. There is much debate surrounding a universal CTC marker and the prognostic implications of this. With the advent of the CellSearch platform, the clinical relevance of EpCAM+ CTCs was demonstrated as is with the studies mentioned by the reviewer. By the same token, there are a number of key papers demonstrating that mesenchymally shifted CTCs are present post therapy which are associated with disease progression (Yu et al., Science Translational Medicine 2013; Li et al., World J Gastroenterology 2015; Satelli et al., Clinical chemistry 2015)
Line 74 (and line 108): The authors should clarify what they intend for “clinically relevant numbers” (see above, please, line 20). Since previously reported data in head and neck cancer have been obtained by different technologies, include, please, the reference(s) and specify, please, the technology used in that case.
(Addressed in response to reviewer 2). In testing CTC enrichment technologies, a number of publications use high cancer cell line spike in numbers (eg 1000’s of cells) to perform recovery studies. Clinically however, patients present with low CTC counts. Therefore, it is desirable to test CTC platforms using ‘clinically-relevant numbers” of spike-in cells. In our study, we demonstrated this using 10-500 cells/ml. Head and neck cancers typically present with low CTC counts (under 50CTCs/ml – Kulasinghe et al., International Journal of Cancer 2015).
Line 75: Amend, please, “phenotyped molecularly” with “characterized molecularly”.
Amended.
Line 86: Since the Material and methods section is at the end of the paper, I suggest inserting here, the definition of CTC clusters and CTMs.
Explained previously.
Line 91: Haematology atlases report the neutrophils (75% of WBC) fairly uniform in size with a diameter between 12 and 15 μm. Why a diameter of 14 μm should discriminate CTCs and CTMs from WBC?
Neutrophils sizes are typically 12-14 µm in diameter. Using this chip, we estimate to have approximately 8% of WBC contamination. Cells larger than 14 µm would be captured by the CTC outlet of the chip which would include CTCs, CTMs and large WBCs (possibly including some neutrophils). The 14 µm cut off separates CTCs and clusters together (as a general population) from the contaminating WBC background.
Line 127 and line 132: How many CTCs are included in CTC clusters and in CTMs? Perform, please, a statistics to compare size of clusters and CTMs (I suggest using non-parametric tests).
The number of CTC clusters per patient blood sample is described in table 1. The size of the clusters and CTMs is shown in Figure 3b. The authors feel that the statistical analysis of CTC clusters in the patient cohorts may be better demonstrated in a larger validation study.
Line 147: Report, please, EGFR status in matched primary tumours, if any data is available. In any case, the authors should briefly comment the translational value of this finding in CTCs.
(Addressed in Reviewer 1 comments). Tumour tissue was not available for this study and this information is not available from pathology as they do not carry out routine molecular characterization of HNC tumour tissue. We utilised the publically available TCGA database to look for the most common alterations. The reason EGFR amplification was assessed was because this has been shown to be present in approximately 15% of HNC tumours (The cancer genome Atlas, TCGA 2015; Comprehensive genomic characterization of head and neck squamous cell carcinomas). Other genes which are amplified include FGFR1 (10%), ERBB2 (5%), DDR2 (3%). Whilst it is desirable to assess other genes, the frequency is low. The reasons for EGFR assessment was to confirm that the CTCs were head and neck cancer in origin.
The translational value has been highlighted in the manuscript (lines 188-189) where the authors mentioned that “Capturing and analyzing CTCs for genetic and molecular alterations allows for the identification of potential targeted therapies which may benefit the patient (e.g. EGFR targeted therapies in HNC).”
Line 169: On what basis the authors state that their microchip preserves CTC clusters better than other technologies? Include here, please, some objective parameters.
In our previous publications where we have used spiral CTC technology, these have been crude processing of the blood samples where blood samples were processed through the spiral chip at 1.7ml/min. Post enrichment, the viability of the cells was low. However, using the straight microfluidic chip, the sample is pumped through the biochip at 0.1ml/min, having an inherently slower and gentler processing. Post enrichment, the viability of the cells appears to be much higher using the straight biochip. The reasons we have not reported on this in this manuscript is that neither platform is sterile for CTC culture in the current form. The authors are in the process of optimising a work flow for this.
Line 275: See, please, above (line 22 and 25).
Addressed above.
Round 2
Reviewer 1 Report
The authors addressed all my review points.